# Diagnostic Performance of Prototype Handheld Ultrasound According to the Fifth Edition of BI-RADS for Breast Ultrasound Compared with Automated Breast Ultrasound among Females with Positive Lumps

**DOI:** 10.3390/diagnostics13061065

**Published:** 2023-03-11

**Authors:** Shahad A. Ibraheem, Rozi Mahmud, Suraini Mohamad Saini, Hasyma Abu Hassan, Aysar Sabah Keiteb

**Affiliations:** 1Department of Radiology, Faculty of Medicine and Health Sciences, Universiti Putra Malaysia, Selangor 43400, Malaysia; 2Center of Diagnostic Nuclear Imaging, Universiti Putra Malaysia, Selangor 43400, Malaysia; 3Department of Radiological Techniques, College of Health and Medical Technologies, Baghdad 10047, Iraq

**Keywords:** prototype handheld ultrasound, BI-RADS, ABUS, breast cancer, HHUS

## Abstract

(1) Objective: To evaluate the diagnostic performance of prototype handheld ultrasound compared to automated breast ultrasound, according to the fifth edition of BI-RADS categorization, among females with positive lumps. (2) Methods: A total of 1004 lesions in 162 participants who underwent both prototype handheld ultrasound and automated breast ultrasound were included. Two radiologists and a sonographer independently evaluated the sonographic features of each lesion according to the fifth BI-RADS edition. The kappa coefficient (κ) was calculated for each BI-RADS descriptor and final assessment category. The cross-tabulation was performed to see whether there were differences between the ABUS and prototype HHUS results. Specificity and sensitivity were evaluated and compared using the McNamar test. (3) Results: ABUS and prototype HHUS observers found the same number of breast lesions in the 324 breasts of the 162 respondents. There was no significant difference in the mean lesion size, with a maximum mean length dimension of 0.48 ± 0.33 cm. The assessment of the lesion’s shape, orientation, margin, echo pattern, posterior acoustic features, and calcification was obtained with good to excellent agreements between ABUS and prototype HHUS observers (κ = 0.70–1.0). There was absolutely no significant difference between ABUS and prototype HHUS in assessment of lesion except for lesion orientation *p* = 0.00. Diagnostic accuracy (99.8% and 97.7–98.9%), sensitivity (99.5% and 98.0–99.0%), specificity (99.8% and 99.6–99.8%), positive predictive value (98.1% and 90.3–96.2%), negative predictive value (90.0% and 84.4–88.7%), and areas under the curve (0.98 and 0.83–0.92; *p* < 0.05) were not significantly different between ABUS and prototype HHUS observers. (4) Conclusion: According to the fifth BI-RADS edition, automated breast ultrasound is not statistically significantly different from prototype handheld ultrasound with regard to interobserver variability and diagnostic performance.

## 1. Introduction

Breast cancer is the number one cancer afflicting women worldwide, including Malaysians. In the latest report by the Malaysian National Cancer Registry, there was an increase in breast cancer incidence from 18,206 between 2007 and 2011 to 21,634 between 2012 and 2016. In addition, the report showed an increasing number of women being diagnosed with breast cancer at younger ages and at more advanced cancer stages compared to the previous report [1]. Moreover, the Malaysian Study on Cancer Survival reported a dismal 66.8% breast cancer survival rate, which is much lower than that reported for other Asian countries such as Japan (96.2%) [2], Korea (92.6%) [3], and Singapore (79.0%) [4]. Significant efforts have been made to improve imaging capabilities to detect breast lesions early. Mammography (MG) remains the gold standard for breast cancer screening, with a sensitivity of 85%. However, the sensitivity of MG screening is limited in women with dense breasts. Ultrasonography (US) is an imaging method that includes diagnostic techniques using ultrasound and is used in several ways concerning breast cancer detection, it is currently regarded as the first line of examination in detecting and characterizing breast lesions, including evaluating breast cancer [5], it is widely employed in characterizing breast masses because it is non-invasive, it enables for real-time visualization, it avoids ionizing radiation, and it is relatively cheap [6]. Nevertheless, it has drawbacks, such as operator-dependency, the time required for examination, and lack of standardization and reproducibility [7]. To overcome these problems, an automated breast ultrasound (ABUS) that is operated using an automated scanner with a larger field of view was recently introduced to provide reproducible, high-resolution images that are not operator-dependent. As reported by a number of prospective studies, by adding ABUS screening to mammography, positive results similar to those associated with HHUS screening were produced, such as reduced rates of interval cancer and increased detection of invasive cancer [8]. To minimize variability in lesion characterization and final lesion assessment from ultrasound, the American College of Radiology developed the breast imaging reporting and data system (BI-RADS) lexicon, which is now in its fifth edition. Furthermore, no substantial differences were discovered between ABUS and HHUS in terms of diagnostic performance for breast cancer despite having technical differences between them [9,10]. Therefore, the purpose of this study was to evaluate the diagnosis performance of prototype HHUS compared to ABUS according to the fifth edition of BI-RADS among females with positive lumps.

## 2. Materials and Methods

### 2.1. Study Participants

A cross-sectional study was conducted between June to October 2021 in the Centre of Diagnostic Nuclear Imaging, Universiti Putra Malaysia. This study was approved by the Ethics Committee of the Ethics Committee for research involving Human Subjects of University Putra Malaysia. Women were randomly and proportionally selected using a computer random number generator. These participants had visited the Centre of Diagnostic Nuclear Imaging for breast pain, had findings on self-palpation, or had a self-motivated request for a breast examination without symptoms. The patients were invited to participate in the study. We excluded patients if they had visible, suspicious signs of breast cancer (such as a palpable mass, bloody nipple discharge, skin redness, skin retraction, or nipple inversion). Patients who had a diagnosis of and were treated for breast cancer or had a history of breast augmentation were excluded. Patients with a larger-than-C-cup size were also excluded. Written informed consent was obtained from all patients. A Google questionnaire was conducted to obtain sociodemographic data and information regarding potential breast cancer risk factors from every patient. Both prototype HHUS and ABUS examinations were conducted for all of the enrolled participants. To be included in the study, the masses identified in prototype HHUS or ABUS images had to be solid tumors or complex cysts. All of the masses were classified as final categories 2, 3, 4, and 5. The American College of Radiology breast imaging reporting and data system (BI-RADS) was used as a guideline to categorize the masses. In total, 162 women, a total of 1004 breast lesions with a mean age of 31.80 ± 9.44 years old and a mean menarche age of 12.35 ± 1.24 years old, age range 19 to 59 years old, were enrolled in this study.

### 2.2. HHUS

The ultrasound examination was performed by one sonographer and two radiologists, with Prototype Handheld Fujikin Ultrasound with a linear transducer 7.5 MHz connected to smartphone, contrast 80, gain 70, and depth 5 cm. Patients were instructed to raise their hands above the head in the supine position. Bilateral breasts tissue was scanned manually in a certain order to ensure that all the breast tissue was covered, and lymph nodes in the armpits were included in the scope of the examination. Suspected masses were observed in two perpendicular scanning planes (longitudinal and axial). The grayscale images were recorded at the same time. All of the images were stored offline. After prototype HHUS examinations, patients underwent ABUS examinations.

### 2.3. ABUS

ABUS scan data were obtained by a sonographer using Invenia ABUS (GE Healthcare, Milwaukee, WI, USA) using an automated 10 MHz linear array transducer automatically applies compression to the breast across the whole breast and obtains images from different views, such as lateral, anteroposterior, and medial (covering areas of 15 cm). The workstation then reconstructs the breast and display 3D volumes in a 2-mm-thick coronal slice from the skin to the chest wall. Patients were placed in a supine position on an examination bed and positioned with the arm above the head. The images were presented on the workstation in any plane after reconstruction with the volume data.

### 2.4. Image Review

HHUS images were interpreted immediately after scanning by one sonographer and two radiologists, and they had more than 5 years of breast imaging experience. Another radiologist, who has more than 3 years of experience of 3D volume ultrasound, reviewed the ABUS images. ABUS assessment was performed on the axial plane as well as the reconstructed coronal plane. All of the sonographers and radiologists are blind to the patient’s identity, the results of other modalities, and the medical background. Each breast unit was assessed according to the terminology of the fifth edition of the BI-RADS lexicon for breast ultrasound [11]: shape (oval, round, or irregular), margin (circumscribed, indistinct, microlobulated, angular, or spiculated), echo pattern (hypoechoic, hyperechoic, isoechoic, anechoic, complex cystic and solid, or heterogeneous), orientation (parallel or perpendicular), posterior features (no posterior features, enhancement, shadowing, or combined pattern), calcification (no calcifications, calcifications in mass, calcifications out of mass, intraductal calcifications), and associated features (architectural distortion, ductal changes, skin changes, or edema). Each observer was instructed to choose the most appropriate BI-RADS descriptor for each lesion. Although there are five categories (from 1 to 5) in the BI-RADS final assessment, the observers were asked to assign lesions to a limited range of BI-RADS categories: 1 (negative), 2 (benign), 3 (probably benign), 4 (suspicious), or 5 (highly suggestive of malignancy). Measurements of lesions detected with prototype HHUS were compared with those obtained using ABUS. The size of the lesion was defined as the maximum diameter via all methods.

### 2.5. Statistical Analysis

Agreement on the BI-RADS lexicon among observers was assessed using kappa statistics. To interpret the kappa coefficients (κ), we used the following definitions: less than 0.20 indicates poor agreement; 0.21–0.40 indicates fair agreement; 0.41–0.60 indicates moderate agreement; 0.61–0.80 indicates good agreement; and 0.81–1.00 indicates very good agreement [12]. The cross-tabulation was performed to see whether there were differences between the ABUS and prototype HHUS results. Receiver operator characteristic (ROC) curve analysis was performed to compare the diagnostic performance of ABUS and prototype HHUS according to selected BI-RADS descriptors. Specifications and sensitivity were evaluated and compared with the McNemar test. According to Leisenring et al., 2000, the calculation and comparison of positive and negative predictive values were done [13]. Statistical analyses were performed using IBM SPSS Software 25.0. The level of significance was set at 0.05.

## 3. Results

### 3.1. Participant Characteristics

A total of 162 females were included in our study, with a mean age 31.80 ± 9.44 years old and a mean menarche age of 12.35 ± 1.24 years old. The majority of respondents (93.2%) were Malay, with nearly two-thirds of them being single (63.6%) and one-fifth of them being married (19.1%). A bachelor’s degree was held by almost one-fifth of the respondents 46.3%, followed by a postgraduate degree 31.5%. Only 2.5% of the respondents earned the highest monthly income in Malaysia, which is RM 12,501. The majority of the respondents had monthly income ranges between less than RM 2500 and RM 2501 to RM 5000 of 27.8% and 37.7%, respectively. Additionally, this study discovered that 39 (24.2%) respondents had a family history of breast cancer. Out of the 39, the majority of respondents (48.7%) said that they had other family members (such as siblings/sisters, aunty, grandmother) who had breast cancer, followed by their mother 43.6% (Table 1).

### 3.2. Diagnostic Performance

#### 3.2.1. ABUS versus Prototype HHUS

A total of 1004 masses from 162 females who underwent prototype HHUS examinations followed by ABUS examinations (including cystic and solid masses) were included in this study. In 151/162 of the responders, the prototype HHUS detected extremely similar lesions to those observed by ABUS (gold standard), which totaled 1004 lesions with a mean length dimension of 0.45 ± 0.32 cm by a length range of 0.20–3.60 cm. In total, 998 lesions (99.4%) were either definitely or probably benign, whereas 4 (0.6%) were malignant as determined by biopsy. Out of 162, only 11 obtained normal results (Table 2). The mean distance from the nipple was 2.00 ± 1.02 cm for the right breast and 1.67 ± 0.71 cm for the left breast. The data showed that there was no statistical difference between the ABUS and prototype HHUS findings for both breasts (*p* > 0.05 in the Chi-square test). For both breasts, the highest agreement was found and was excellent for ABUS and prototype HHUS observers (HHUS observers 1, 2, and 3; κ = 0.95, κ = 0.95, and κ = 0.97), respectively.

#### 3.2.2. Interobserver Agreement of ABUS and HHUS in the Fifth Edition of BI-RADS Descriptors and Final Assessment

Shape of the Lesion

According to the 4th Table in Section 3.2.2 findings, there was no difference between the ABUS and prototype HHUS findings for both breasts that could be not considered statistically significant (*p* > 0.05). Regarding the shape of the lesion, both modalities show agreement revealed that for the right breast, 361 were rounded, 196 were elliptical, and 2 were irregular, whereas for the left breast, 300 were rounded, 144 were elliptical, and 1 was irregular. The relation between both modalities regarding the shape is shown in Table 3.

2.Orientation of the lesion

Regarding the lesion’s orientation between the ABUS and prototype HHUS revealed agreement in 555 parallel and 3 perpendicular lesions for the right breast and 443 parallel and 2 perpendicular lesions for the left breast, and there was a significant difference between the ABUS and prototype HHUS findings for both breasts, with a *p*-value of 0.05 (Table 4).

3.Margin of the lesion

The margin of the lesion both modalities showed agreement in 546 circumscribed, 11 microlobulated, 1 angular, and 1 spiculated for the right breast, compared to 435 circumscribed, 3 angular, 6 microlobulated, and 1 spiculated for the left breast; on the other hand, there was no statistically significant difference between the results of the ABUS and prototype HHUS for both breasts *p* > 0.05 (Table 5).

4.Echo pattern of the lesion

Table 6’s findings indicate that there was no statistically significant difference between the ABUS and HHUS findings for each breast, *p* > 0.05. For both breasts, with regards to the echo pattern of the lesion, ABUS and prototype HHUS proved agreement using the chi square: 524 were anechoic, 0 were isoechoic, 19 were hypoechoic, 10 were hyperechoic, 1 was complex cystic and solid, and 5 were heterogenous for the right breast, whereas 420 were anechoic, 17 were hypoechoic, 6 were hyperechoic, 2 were heterogenous, and there no findings of isoechoic for the left breast.

5.Posterior Acoustic Features of the lesion

The posterior acoustic features of the lesions were compared at baseline between the ABUS and prototype HHUS findings, indicating that there was no statistically significant difference between the results from the prototype HHUS and the ABUS by either breast, *p* > 0.05. Regarding the posterior acoustic features of the lesion, both modalities indicated agreement in 465 enhancements, 34 shadowings, and no findings for combined and absent features for the right breast, while 397 enhancements, 28 shadowings, 1 combined feature, and 1 absent feature were found for the left breast (Table 7).

6.Calcification of the lesion

According to the findings in Table 8, there was no significant difference between the results of the prototype HHUS and the ABUS for both breasts *p* > 0.05. According to both modalities, 542 of the lesions had no calcifications, 10 had calcifications in the mass, 7 had calcifications outside of the mass, and there were no indications for intraductal calcifications in the right breast. Nevertheless, there were 433 cases of no calcifications, 7 cases of calcifications in masses, 5 cases of calcifications outside of masses, and no observations of intraductal calcifications in the left breast.

7.BI-RADS Assessment of The Lesion

According to Table 9 findings, there was no statistically significant difference between the results from the prototype HHUS and the ABUS for either breast, *p* > 0.05, regarding the BI-RADS assessment of the lesion. ABUS and prototype HHUS revealed agreement as follows: 522 were BI-RADS 2/benign, 31 were BI-RADS 3/probably benign, and 6 were BI-RADS 4/suspicious for right breast, whereas 420 were BI-RADS 2/benign, 22 were BI-RADS 3/probably benign, and 3 were BI-RADS 4/suspicious for the left breast. There were no findings in terms of BI-RADS 1/normal and BI-RADS 5/highly suggestive of malignancy for both breasts.

Table 10 summarizes the interobserver agreement of BI-RADS descriptors and the final assessment category in ABUS and prototype HHUS observers; the overall agreement for all descriptors and final assessment category was good to excellent. There were no significant differences in κ values of BI-RADS lexicons between ABUS and prototype HHUS except for orientation (ABUS except for associated features (Table 1, the 2nd Figure in Section 3.3)).

8.Location of the Lesion

According to Table 11 findings, there was no significant difference between ABUS and prototype HHUS findings for both breasts (*p* > 0.05). The maximum agreement was found for location, and excellent agreement was found for ABUS with different prototype HHUS observers (κ = 0.92–0.96) for both breasts. Regarding the lesion’s site, both modalities revealed agreement in 147 cases where right upper inner quadrant (RUIQ), right lower inner quadrant (RLIQ), right lower outer quadrant (RLOQ), and right upper outer quadrant (RUOQ) were allocated to the right breast, and in 139 cases were left upper outer quadrant (LUOQ), left lower outer quadrant (LLOQ), left lower inner quadrant (LLIQ), and left lower inner quadrant (LUIQ) were allocated to the left breast. In UOQ, both breasts had the most lesions. * Significant result (*p* < 0.05).

9.Location of the Lesion

According to Table 11 findings, there was no significant difference between ABUS and prototype HHUS findings for both breasts (*p* > 0.05). The maximum agreement was found for location, and excellent agreement was found for ABUS with different prototype HHUS observers (κ = 0.92–0.96) for both breasts. Regarding the lesion’s site, both modalities revealed agreement in 147 cases where right upper inner quadrant (RUIQ), right lower inner quadrant (RLIQ), right lower outer quadrant (RLOQ), and right upper outer quadrant (RUOQ) were allocated to the right breast, and in 139 cases where left upper outer quadrant (LUOQ), left lower outer quadrant (LLOQ), left lower inner quadrant (LLIQ), and left lower inner quadrant (LUIQ) were allocated to the left breast. In UOQ, both breasts had the most lesions. * Significant result (*p* < 0.05).

### 3.3. Sensitivity, Specificity, and Receiver Operating Characteristic Analysis

There was no significant difference in the overall diagnostic accuracy of ABUS and prototype HHUS observers (*p* > 0.05) for the 1004 masses. The sensitivity and specificity of ABUS were 99% and 97%. The PPV and NPV of ABUS were 96% and 94%. The sensitivity and specificity with different prototype HHUS observers 1 through 3 were 97%, 97%, 98% and 90%, 93%, 96%, respectively. The PPV and NPV with different HHUS observers 1, 2, and 3 were 80%, 90%, 94%, and 73%, 73%, 76%, respectively. The AUCs were 0.99 for ABUS and 0.98 for HHUS observer 1, 0.97 for HHUS observer 2, and 0.97 for HHUS observer 3. Table 12 is provided as an assessment of the methods’ accuracy (Figure 1, Figure 2 and Figure 3).

## 4. Discussion

The main goal of this study was to assess the diagnostic accuracy of automated breast ultrasound compared to handheld ultrasound in lesions detection, description, and interpretation. The fifth edition of BI-RADS was used to describe the morphological characteristics of each lesion, and the final BI-RADS classification for ABUS and HHUS was determined. In evaluating the results of a screening program, we observed that ABUS and prototype HHUS both detected a comparable number of breast lesions. Previous research showed that ABUS findings were significantly more accurate in measuring breast lesions’ sizes compared to HHUS results. In 151/162 of the respondents, ABUS (gold standard) and prototype HHUS detected 1004 lesions with a maximum mean length dimension of 0.48 ± 0.33 cm with length range (0.20–3.60) cm. Additionally, the results of the Chen et al. (2021) study are higher than our study; the mean lesion diameters detected with the ABUS and HHUS were 23.2 ± 9.5 mm and 22.3 ± 6.3 mm, respectively [14]. The mean lesion size between ABUS and prototype HHUS observers did not differ significantly. The size of the masses may have an impact on the inter-observer agreement. Our studies found that for masses, the overall perfect agreement for size was at the same level. However, Shin et al. (2011) showed significant inter-observer agreement for masses greater than or equal to 7 cm (k = 0.750) and fair inter-observer agreement for masses less than or equal to 7 cm (k = 0.350) [15]. In our investigation, the two imaging modalities revealed that the benign lesions ranged in size from 0.2 to 2.5 cm; most of benign lesion were cysts followed by fibrocystic and fibroadenoma, whereas the malignant lesions ranged in size from 0.6 to 3.6 cm. According to Chen et al.’s (2013) study, the mean size of benign and malignant lesions was greater (1.97 cm and 1.76 cm, respectively) by ABUS and HHUS [16]. Chang et al. (2010) published a different study with smaller malignant and benign lesions, with mean sizes of 1.55 cm and 1.35 cm, respectively [17]. As shown in a systematic review study conducted by Ibraheem et al. in 2022, the identified malignancies had a mean percentage of 94% (81–100%) in comparison to the non-cancer in all studies, and the found tumors had a mean size of 2.1 cm. The data in the literature indicates that ABUS and HHUS perform similarly to one another in terms of differentiating between malignant and benign breast tumors [10].

Our study’s findings revealed that the use of BI-RADS descriptors and final assessment categories was accompanied by a comparatively high level of intra-observer agreement. Regarding the evaluation of lesions’ shape, orientation, margin, echo pattern, posterior acoustic features, and calcification of lesions, there were good to excellent agreements between ABUS and HHUS observers. There were no significant differences between the three observers’ diagnostic performances, which were all good to excellent. However, HHUS observer 3 discovered a moderate agreement level for the lesion orientation on the left breast. Round, elliptical, and irregular shapes all fall under the category of shape, which is the outline of a mass. It is one of the most significant morphological features for differentiating benign masses from malignant tumors [13,18]. Particularly, malignant breast tumors on ABUS are independently correlated with irregular shape [11]. Previous ABUS analysis showed that irregular shapes had a 52.1% positive predictive value (PPV), 95.0% sensitivity, and 66.2% accuracy [19,20]. In our study, the overall agreement regarding the shape of the lesion was excellent for ABUS with different HHUS observers. However, we observed higher interobserver agreement than [6,11,21]. Significant agreement on orientation was obtained by Shin et al. (2011) for both small masses (<0.7 cm) (k = 0.68) and large masses (>0.7 cm) (k = 0.72) [15]. On the other hand, there was only moderate agreement on orientation for small masses (1.0 cm) (k = 0.530) and substantial agreement for large masses (>1.0 cm) (k = 0.634). In our investigation, the interobserver agreement for ABUS with several HHUS observers ranged from good to moderate. This difference might be the result of the ultrasound probe applying different pressures to the mass, which could produce deformations and affect the length-to-width diameter ratio [22]. Nevertheless, Liu et al. (2022) evaluated a series of 331 histopathologically verified lesions and observed a perfect interobserver agreement for circumscribed margins between ABUS and HHUS, while they showed fair to a moderate agreement for the margins of indistinct (k = 0.38), angular (k = 0.37), microlobulated (k = 0.46), and spiculated (k = 0.31) lesions [21]. As per result of our findings, Choi et al. (2018) observed good interobserver margin agreement [11].

Our findings indicated that ABUS and several HHUS observers had perfect agreement in terms of echo patterns (k = 0.92–0.97). In accordance with our observation, Liu et al. (2022) found perfect agreement for anechoic echo pattern assessment (k = 0.918); good agreement for hyperechoic pattern assessment (k = 0.700), hypoechoic pattern assessment (k = 0.676), and isoechoic pattern assessment (k = 0.614); and moderate agreement for complex cystic and solid pattern assessment (k = 0.511) and heterogeneous pattern assessment (k = 0.527) [21]. Choi et al. (2018) observed excellent agreement for anechoic echo pattern evaluation (k = 0.92); significant agreement for hypoechoic (k = 0.60) and isoechoic (k = 0.64); moderate agreement for hyperechoic (k = 0.47), complicated cystic and solid (k = 0.44), and heterogeneous (k = 0.527); and reasonable agreement for hyperechoic (k = 0.37) echo pattern assessment [11].

Vourtsis and Kachulis (2018) evaluated 1665 women and observed perfect interobserver agreement between ABUS and HHUS (k = 0.85–0.95), which is less than our findings in terms of posterior features [9]. Liu et al. (2022) observed perfect agreement (k = 0.779), which is consistent with our findings [21]. The findings of our investigation showed good to excellent agreement between ABUS and different prototype HHUS observers. On the other hand, our results showed higher interobserver agreement than that which Liu et al. (2022) and Choi et al. (2018) observed [11,21]. Due to the small number of lesions with outside calcification or intraductal calcification, Liu et al. (2022) used dichotomized categories for presence or absence calcifications instead of descriptors from the fifth edition of BI-RADS. The results showed substantial agreement for these dichotomized categories’ assessment of calcification (k = 0.604), which was higher than that of Choi et al. (2018), who also used dichotomized different study samples, which may account for this discrepancy [11,21].

According to the findings of our study, there was a relatively high level of agreement in the use of BI-RADS final assessment categories for ABUS with different HHUS observers (k = 0.84–0.92). Liu et al. (2022) observed perfect inter-observer agreement for categories 2 (κ = 0.918), 3 (κ = 0.870), 4 (κ = 0.837), 5 (κ = 0.855), and overall (κ = 0.752), which was better than the outcomes of previous studies [11,23,24]. The inter-observer agreement in BI-RADS classification between the two assessors was excellent (99.8%, κ = 0.996) in a similar study by Vourtsis and Kachulis (2018) [9]. Jia et al. (2020) observed significant agreement (0.94) between ABUS and HHUS (κ = 0.85, 95% CI 0.81–0.89, *p* <0.01) in characterizing the radiological findings, with a BI-RADS total score in all women with mammographic dense breasts [25].

Our results do not agree well with published data from prior studies in terms of sensitivity, specificity, diagnostic accuracy, positive predictive value (PPV), and negative predictive value (NPV). Chen et al. (2013) evaluated 152 benign and 67 malignant breast lesions and reported a sensitivity of 92.5% vs. 88.0% (ABUS vs. HHUS), a specificity of 86.2% vs. 87.5%, and diagnostic accuracy of 88.1% vs. 87.2% [16]. Wang et al. (2012) investigated 239 breast tumors (85 malignant and 154 benign) and discovered that, for ABUS vs. HHUS, there was sensitivity of 95.3% vs. 90.6%, specificity of 80.5% vs. 82.5%, diagnostic accuracy of 85.8% vs. 85.%, a PPV of 73.0% vs. 74.0%, and an NPV of 93.3% vs. 94.1% [26]. A significant difference between ABUS and HHUS was found in sensitivity (92.23% vs. 81.55%; *p* = 0.007); nevertheless, there was no significant difference in terms of specificity (77.62% vs. 80.04%), PPV (46.12% vs. 45.90%), NPV (95.43% versus 97.96%), or accuracy (80.63% vs. 80.13%; *p* > 0.05 for all). However, in Niu et al. (2019), a significant difference was found between ABUS and HHUS in terms of specificity (86.2% vs. 87.5%; *p* = 0.018) and diagnostic accuracy (97.70% vs. 96.54%; *p* = 0.022) [27], but there was no difference in terms of sensitivity (77.78% vs. 62.50%; *p* >0.05) in a report by Choi et al. (2018) using a large sample (1866 ABUS and 3700 HHUS participants among 5566 women) [11]. According to the literature review conducted by Ibraheem et al. (2022), ABUS had considerably greater diagnostic accuracy, positive predictive value, and negative predictive value in our study. The radiologists’ performance in terms of detection, sensitivity, and specificity did not significantly differ between the two modalities (*p* > 0.05) [10].

An AUC of 0.91 to 0.93 for ABUS and 0.83 to 91 for HHUS, with no significant difference between ABUS and HHUS, was observed in a prior study utilizing the fifth edition of BI-RADS [21,28,29]. Our study predicted an AUC of 0.88 to 1.0, which is comparable to the range of the other studies [30,31,32,33,34]. In our analysis, ABUS and HHUS did not miss any malignant lesions. Compared to HHUS, more benign lesions were found. Focal fat lobules were most likely the cause of a few of the solid nodules found on ABUS. Although we did not compare the number of cysts in this investigation, ABUS appeared to identify more cysts than HHUS. Cysts typically showed up on ABUS as being more hypoechoic, and some of them resembled complicated cysts.

According to the earlier study by Schmachtenberg et al. (2017), agreement regarding lesion localization was 0.94 for ABUS and MRI and 0.91 for HHUS and MRI. The three observers’ estimates of agreement about lesion localization (same quadrant) ranged from 0.92 to 0.96 [35]. Additionally, Chae et al. (2013) demonstrated great reproducibility for lesion site and distance from the nipple, which is equivalent to our result [36]. Contrary to our findings, a number of studies have assessed the inter-observer reliability of location, the distance from the nipple as well as from the skin, and the size of the lesion with less inter-observer reliability assessment [6,11,37,38,39]. Wang et al. (2012) found similar results to ours, with the mean distance from the nipple identified by ABUS among 30 lesions measuring 3.01 ± 1.54 cm and HHUS among 27 lesions measuring 3.36 ± 1.72 cm [26].

Even though the present study has several advantages, the accuracy of the prototype HHUS will also help doctors and other medical professionals use it in vital situations where time is of essence (emergency room, intensive care unit), or when the location prefers the use of HHUS devices (remote location, doctor’s office). This is one of the study’s main strengths. Therefore, by encouraging and supporting early breast cancer screening, this study will assist Malaysian women, although some limitations must be addressed. Firstly, the design for phase one is cross-sectional; there are no follow-up data for women who are determined to be non-cases. Due to verification bias, this might cause an overestimation of the test sensitivity. However, women with potentially benign findings need to have them confirmed by MRI or biopsy; as a result, the likelihood of false negatives should be relatively low. Second, lesions might be missed on ABUS if they have peripheral location. This technical drawback reduces the diagnostic performance of the method comparing to HHUS, especially in larger breasts, and could represent a cause for the misdiagnosis of cancer. The main limitation of ABUS is its inability to assess the axilla, the area behind the nipple. Furthermore, the results could not apply to all Malaysian women because of the limited sample size, the majority of respondents being of Malay ethnicity, and the single site study. Therefore, further research is required to reproduce the findings using a bigger sample size. The price of ABUS and the fact that Malaysia does not offer ABUS screening unless a reference is received are additional factors. As a result, fewer women may have had access to screening due to systemic barriers. There are some recommendations based on the results of the current study and the limitations that were noted. Initially, we recommend the use of HHUS as an adjacent device to mammogram, and MRI for dense breasts. Likewise, we recommended the use of HHUS as an adjacent device to ABUS in order to cover the axilla and area behind the nipple. For future study, the use of HHUS as a breast self-device by women is advised since that was the main objective of our study, and because of pandemic we could not achieve that objective.

## 5. Conclusions

This study revealed that the baseline screening uptake among Malaysian females accurately represents the current screening for prototype HHUS and ABUS. We discovered that a similar number of breast lesions were diagnosed by ABUS compared to prototype HHUS out of the overall sample. According to these findings, prototype HHUS is not statistically significantly different from ABUS in terms of interobserver variability or diagnostic performance.

## Figures and Tables

**Figure 1 diagnostics-13-01065-f001:**
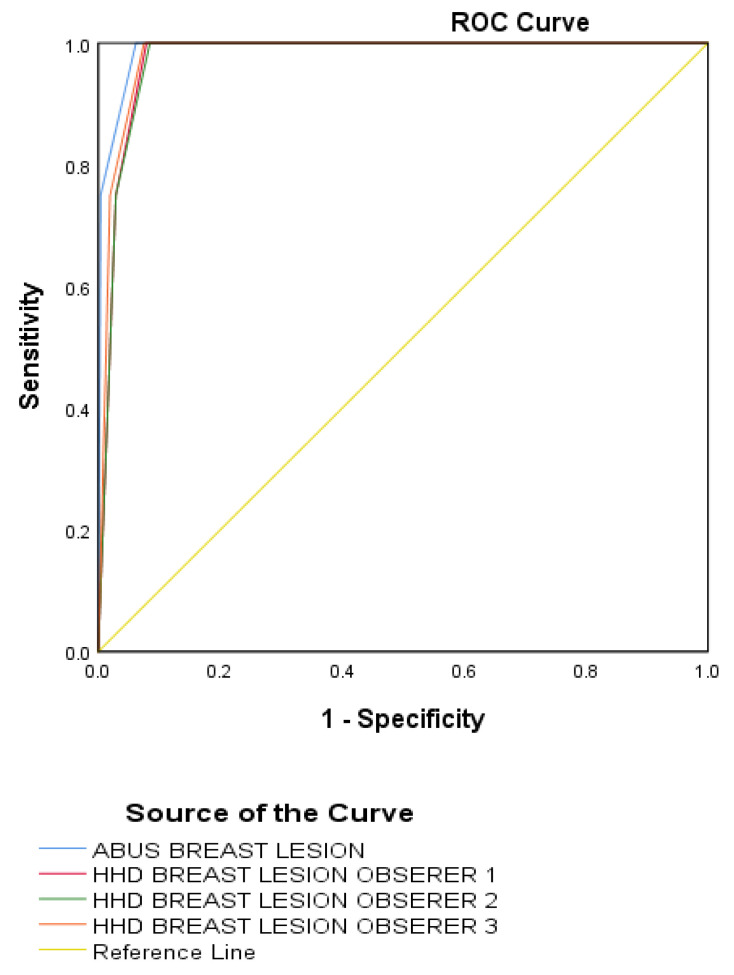
Receiver operating characteristic curves of ABUS and prototype HHUS observers for discerning breast lesions between non-malignant and malignant lesions. ABUS = automated breast ultrasound system; HHUS = hand-held ultrasound; and ROC = receiver operating characteristic.

**Figure 2 diagnostics-13-01065-f002:**
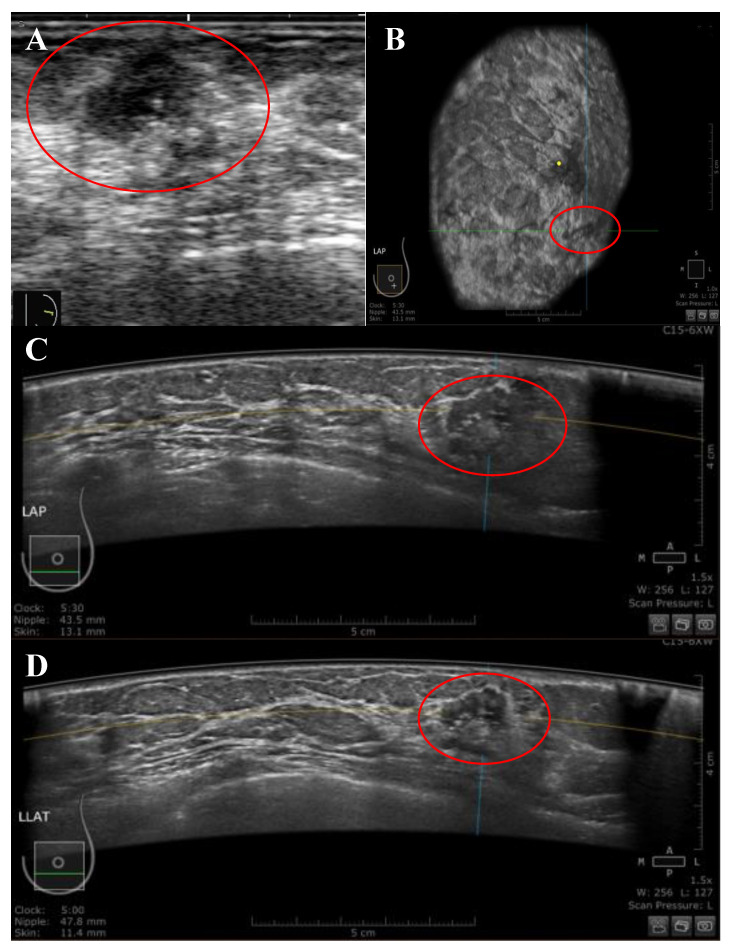
A 51-year-old Malay female patient presented with a left LOQ breast lump at a distance of 43.5 mm from the nipple. (**A**) Handheld US image showing irregular-shaped, hypoechoic, non-parallel mass, and heterogenous echo pattern with spiculated margins and acoustic shadowing also showed diffuse inside and outside mass calcifications. The mass was classified as BIRADS category 4 by HHUS. (**B**–**D**) the mass was presented by ABUS in three orthogonal planes: coronal (**B**), transverse (**C**), and sagittal (**D**). The mass was classified as BIRADS category 4 by ABUS. The histological evidence indicated a malignant lesion. The yellow dot marks the position of the nipple, and the red circle marks the lesion.

**Figure 3 diagnostics-13-01065-f003:**
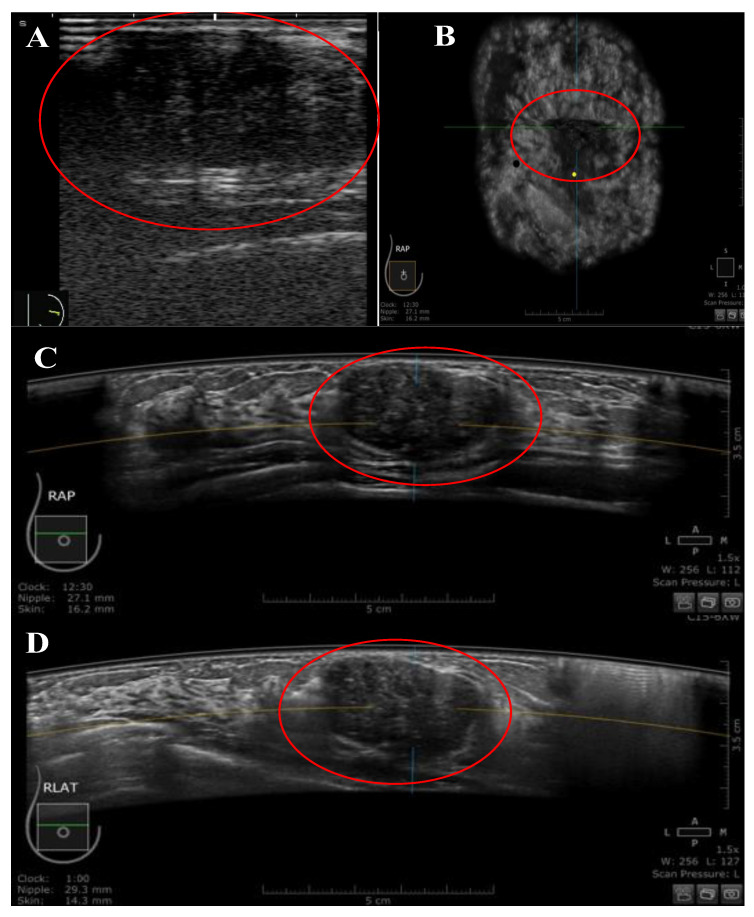
A 29-year-old Malay female patient was presented with a right UIQ breast lump at a distance of 27.1 mm from the nipple. (**A**) Handheld US image showing an oval-shaped, hypoechoic, parallel mass, and heterogenous echo pattern with regular margins. The mass was classified as BIRADS category 3 by HHUS. (**B**–**D**) the mass was presented by ABUS in three orthogonal planes: coronal (**B**), transverse (**C**), and sagittal (**D**). The mass was classified as BIRADS category 3 by ABUS. The yellow dot marks the position of the nipple, and the red circle marks the lesion.

**Table 1 diagnostics-13-01065-t001:** Characteristics of socio-demographic and other factors at baseline (*n* = 162).

Variables	Mean ± SD	Frequency *N*	Percentage %
**Age (years)** (min–max)	31.80 ± 9.44 (19–59)		
**Race**			
Malay		151	93.2
Chinese		3	1.9
Indian		4	2.5
Others		4	2.5
**Marital status**			
Single		101	62.3
Married		52	32.1
Divorced/separated		6	3.7
Widowed		3	1.9
**Education level**			
Primary/secondary		9	5.6
Diploma		26	16.0
Degree		76	46.9
Postgraduate		51	31.5
**Occupation**			
Student		23	14.2
Working		120	74.1
Housewife		17	10.5
Retired		2	1.2
**Income (RM)**			
Less than RM 2500		45	27.8
RM 2501 to RM 5000		61	37.7
RM 5001 to RM 7500		23	14.2
RM 7501 to RM 10,000		16	9.9
RM 10,001 to RM 12,500		13	8.0
RM 12,501 and above		4	2.5
**Family history of BC**			
Yes		39	24.1
No		123	75.9
**Family members**			
Mother		17	43.6
Sister		3	7.7
Others		19	48.7
**Menarche age (years)**	12.35 ± 1.24		
(min–max)	(8–16)		

**Table 2 diagnostics-13-01065-t002:** HHUS observers results compared with ABUS (gold standard).

HHUS Observers	ABUS (Gold Standard)				*p* Value
		Malignant	Non-Malignant	Total	
**Observer 1**	**Malignant**	10	0	10	
	**Non-Malignant**	4	960	964	0.072
	**Total**	14	960	974	
**Observer 2**	**Malignant**	9	0	9	0.053
	**Non-Malignant**	3	961	964	
	**Total**	12	961	973	
**Observer 3**	**Malignant**	7	0	7	0.059
	**Non-Malignant**	1	972	973	
	**Total**	8	972	980	

**Table 3 diagnostics-13-01065-t003:** HHUS observers compared with ABUS (gold standard) in terms of lesion shape.

HHUS Observers	ABUS (Gold Standard)												
Right Breast		Rounded	Elliptical	Irregular	Total	*p* Value	Left Breast		Rounded	Elliptical	Irregular	Total	*p* Value
**Observer 1**	**Rounded**	329	11	0	340	0.12	**Observer 1**	**Rounded**	286	5	0	291	1.01
	**Elliptical**	14	180	0	194			**Elliptical**	4	142	0	146	
	**Irregular**	0	0	2	2			**Irregular**	0	0	1	1	
	**Total**	343	191	2	536			**Total**	290	147	1	438	
**Observer 2**	**Rounded**	339	7	0	346	0.1	**Observer 2**	**Rounded**	286	3	0	289	0.14
	**Elliptical**	7	183	0	190			**Elliptical**	1	144	0	145	
	**Irregular**	0	0	2	2			**Irregular**	0	0	1	1	
	**Total**	346	190	2	538			**Total**	287	147	1	435	
**Observer 3**	**Rounded**	338	6	0	344	0.05	**Observer 3**	**Rounded**	289	5	0	294	0.12
	**Elliptical**	9	185	0	194			**Elliptical**	1	144	0	145	
	**Irregular**	0	0	2	2			**Irregular**	0	0	1	1	
	**Total**	347	191	2	540			**Total**	290	149	1	440	

**Table 4 diagnostics-13-01065-t004:** HHUS observers compared with ABUS (gold standard) in terms of lesion orientation.

HHUS Observers	ABUS (Gold Standard)										
Right Breast		Parallel	Not Parallel	Total	*p* Value	Left Breast		Parallel	Not Parallel	Total	*p* Value
**Observer 1**	**Parallel**	533	1	534	0.000 *	**Observer 1**	**Parallel**	436	1	437	0.002 *
	**Not Parallel**	0	2	2			**Not Parallel**	0	1	1	
	**Total**	533	3	536			**Total**	436	2	438	
**Observer 2**	**Parallel**	536	1	537	0.000 *	**Observer 2**	**Parallel**	433	1	434	0.002 *
	**Not Parallel**	0	2	2			**Not Parallel**	0	1	1	
	**Total**	536	3	539			**Total**	433	2	435	
**Observer 3**	**Parallel**	538	1	539	0.000 *	**Observer 3**	**Parallel**	438	1	439	0.002 *
	**Not Parallel**	0	2	2			**Not Parallel**	0	1	1	
	**Total**	538	3	541			**Total**	438	2	440	

* Significant result (*p* < 0.05).

**Table 5 diagnostics-13-01065-t005:** HHUS observers compared with ABUS (gold standard) in terms of lesion margin.

HHUS Observers	ABUS (Gold Standard)							
		Circumscribed	Indistinct	Angular	Microlobulated	Spiculated	Total	*p* Value
**Right Breast**								
**Observer 1**	**Circumscribed**	520	0	-	0	0	520	0.07
	**Indistinct**	0	1	-	0	0	1	
	**Angular**	-	0	-	-	-	-	
	**Microlobulated**	3	0	-	11	0	14	
	**Spiculated**	0	0	-	0	1	1	
	**Total**	523	1	-	11	1	536	
**Observer 2**	**Circumscribed**	521	0	-	0	0	521	0.09
	**Indistinct**	0	1	-	0	0	1	
	**Angular**	-	-	-	-	-	-	
	**Microlobulated**	5	0	-	11	0	16	
	**Spiculated**	0	0	-	0	1	1	
	**Total**	526	1	-	11	1	539	
**Observer 3**	**Circumscribed**	526	0	-	0	0	526	0.09
	**Indistinct**	0	1	-	0	0	1	
	**Angular**	-	-	-	-	-	-	
	**Microlobulated**	2	0	-	11	0	13	
	**Spiculated**	0	0	-	0	1	1	
	**Total**	528	1	-	11	1	541	
**Left Breast**								
**Observer 1**	**Circumscribed**	425	-	1	0	0	426	1
	**Indistinct**	-	-	-	-	-	-	
	**Angular**	0	-	2	0	0	2	
	**Microlobulated**	2	-	0	6	1	9	
	**Spiculated**	0	-	0	0	1	1	
	**Total**	427	-	3	6	2	438	
**Observer 2**	**Circumscribed**	420	-	1	0	1	426	1.06
	**Indistinct**	-	-	-	-	-	-	
	**Angular**	0	-	2	0	0	2	
	**Microlobulated**	4	-	0	6	0	10	
	**Spiculated**	-	-	0	0	1	1	
	**Total**	424	-	3	6	2	435	
**Observer 3**	**Circumscribed**	427	-	1	0	1	429	0.99
	**Indistinct**	-	-	-	-	-	-	
	**Angular**	0	-	2	0	0	2	
	**Microlobulated**	2	-	0	6	0	8	
	**Spiculated**	0	-	0	0	1	1	
	**Total**	429	-	3	6	2	440	

**Table 6 diagnostics-13-01065-t006:** HHUS observers compared with ABUS (gold standard) in terms of echo pattern.

HHUS Observers	ABUS (Gold Standard)								
		Anechoic	Isoechoic	Hypoechoic	Hyperechoic	Complex Cystic and Solid	Heterogenous	Total	*p* Value
**Right Breast**									
**Observer 1**	**Anechoic**	495	-	0	0	0	0	495	1.22
	**Isoechoic**	-	-	-	-	-	-	-	
	**Hypoechoic**	6	-	19	0	0	0	25	
	**Hyperechoic**	0	-	0	10	0	0	10	
	**Complex**	0	-	0	0	1	0	1	
	**Cystic and Solid**								
	**Heterogenous**	0	-	0	0	0	5	5	
	**Total**	501	-	19	10	1	5	536	
**Observer 2**	**Anechoic**	499	-	0	0	0	0	499	1.03
	**Isoechoic**	-	-	-	-	-	-	-	
	**Hypoechoic**	5	-	19	0	0	0	24	
	**Hyperechoic**	0	-	0	10	0	0	11	
	**Complex**	0	-	0	0	1	0	1	
	**Cystic and Solid**								
	**Heterogenous**	0	-	0	0	0	5	5	
	**Total**	504	-	19	10	1	5	539	
**Observer 3**	**Anechoic**	502	-	1	0	0	0	502	1.14
	**Isoechoic**	-	-	-	-	-	-	-	
	**Hypoechoic**	4	-	19	0	0	0	23	
	**Hyperechoic**	0	-	0	11	0	0	11	
	**Complex**	0	-	0	0	1	0	1	
	**Cystic and Solid**								
	**Heterogenous**	0	-	0	0	0	5	5	
	**Total**	502	-	19	11	1	5	541	
**Left Breast**									
**Observer 1**	**Anechoic**	408	-	0	0	-	0	408	0.06
	**Isoechoic**	-	-	-	-	-	-	-	
	**Hypoechoic**	5	-	17	0	-	0	22	
	**Hyperechoic**	0	-	0	6	-	0	6	
	**Complex**	0	-	0	0	-	-	0	
	**Cystic and Solid**								
	**Heterogenous**	-	-	-	-	-	2	2	
	**Total**	413	-	17	6	-	2	438	
**Observer 2**	**Anechoic**	407	-	0	0	-	0	407	0.34
	**Isoechoic**	-	-	-	-	-	-	-	
	**Hypoechoic**	3	-	17	0	-	0	20	
	**Hyperechoic**	0	-	0	6	-	0	6	
	**Complex**	-	-	-	-	-	-	-	
	**Cystic and Solid**								
	**Heterogenous**	0	-	0	0	-	2	2	
	**Total**	410	-	17	6	-	2	435	
**Observer 3**	**Anechoic**	412	-	0	0	-	0	412	0.33
	**Isoechoic**	-	-	-	-	-	-	-	
	**Hypoechoic**	3	-	17	0	-	0	20	
	**Hyperechoic**	0	-	0	6	-	0	6	
	**Complex**	-	-	-	-	-	-	-	
	**Cystic and Solid**								
	**Heterogenous**	0	-	0	0	-	2	2	
	**Total**	415	-	16	6	-	2	440	

**Table 7 diagnostics-13-01065-t007:** HHUS observers compared with ABUS (gold standard) in terms of posterior acoustic features.

HHUS Observers	ABUS (Gold Standard)						
		Enhancement	Shadowing	Combined	Absent	Total	*p* Value
**Right Breast**							
**Observer 1**	**Enhancement**	493	0	-	-	493	0.21
	**Shadowing**	0	33	-	-	33	
	**Combined**	-	-	-	-	-	
	**Absent**	10	0	-	-	10	
	**Total**	503	33	-	-	536	
**Observer 2**	**Enhancement**	495	0	-	-	495	0.93
	**Shadowing**	0	33	-	-	33	
	**Combined**	-	-	-	-	-	
	**Absent**	11	0	-	-	11	
	**Total**	506	33	-	-	539	
**Observer 3**	**Enhancement**	497	0	-	-	497	0.97
	**Shadowing**	0	33	-	-	33	
	**Combined**	-	-	-	-	-	
	**Absent**	11	0	-	-	11	
	**Total**	508	33	-	-	541	
**Left Breast**							
**Observer 1**	**Enhancement**	407	0	0	-	407	0.06
	**Shadowing**	0	24	0	-	24	
	**Combined**	0	0	1	-	1	
	**Absent**	6	0	0	-	6	
	**Total**	413	24	1	-	438	
**Observer 2**	**Enhancement**	407	0	0	-	407	0.51
	**Shadowing**	0	24	0	-	24	
	**Combined**	0	0	1	-	1	
	**Absent**	3	0	0	-	-	
	**Total**	410	24	1	-	435	
**Observer 3**	**Enhancement**	411	0	0	-	411	0.06
	**Shadowing**	0	24	0	-	24	
	**Combined**	0	0	1	-	1	
	**Absent**	4	0	0	-	4	
	**Total**	415	24	1	-	440	

**Table 8 diagnostics-13-01065-t008:** HHUS observers compared with ABUS (gold standard) in terms of lesions calcification.

HHUS Observers	ABUS (Gold Standard)						
		No Calcifications	Calcifications in Mass	Calcifications Out of Mass	Intraductal Calcifications	Total	*p* Value
**Right Breast**							
**Observer 1**	**No Calcifications**	514	2	2	-	518	0.87
	**Calcifications in Mass**	0	8	0	-	8	
	**Calcifications Out of Mass**	5	0	5	-	10	
	**Intraductal Calcifications**	-	-	-	-	-	
	**Total**	519	10	7	-	536	
**Observer 2**	**No Calcifications**	522	2	3	-	527	0.82
	**Calcifications in Mass**	0	7	1	-	8	
	**Calcifications Out of Mass**	1	0	3	-	4	
	**Intraductal Calcifications**	-	-	-	-	-	
	**Total**	523	9	7	-	539	
**Observer 3**	**No Calcifications**	523	2	3	-	528	0.91
	**Calcifications in Mass**	0	8	0	-	8	
	**Calcifications Out of Mass**	1	0	4	-	5	
	**Intraductal Calcifications**	-	-	-	-	-	
	**Total**	493	10	7	-	541	
**Left Breast**							
**Observer 1**	**No Calcifications**	423	1	1	-	425	0.43
	**Calcifications in Mass**	2	6	0	-	8	
	**Calcifications Out of Mass**	1	0	4	-	5	
	**Intraductal Calcifications**	-	-	-	-	-	
	**Total**	426	7	5	-	438	
**Observer 2**	**No Calcifications**	419	1	1	-	421	0.43
	**Calcifications in Mass**	4	6	0	-	10	
	**Calcifications Out of Mass**	0	0	4	-	4	
	**Intraductal Calcifications**	-	-	-	-	-	
	**Total**	423	7	5	-	435	
**Observer 3**	**No Calcifications**	425	1	1	-	427	0.37
	**Calcifications in Mass**	3	6	0	-	9	
	**Calcifications Out of Mass**	0	0	4	-	4	
	**Intraductal Calcifications**	-	-	-	-	-	
	**Total**	428	7	5	-	440	

**Table 9 diagnostics-13-01065-t009:** HHUS observers compared with ABUS (gold standard) in terms of BIRADS assessment.

HHUS Observers	ABUS (Gold Standard)							
		BIRADS1	BIRADS2	BIRADS3	BIRADS4	BIRADS5	Total	*p* Value
**Right Breast**								
**Observer 1**	**BIRADS1**	-	-	-	-	-	-	0.23
	**BIRADS2**	-	490	2	0	-	492	
	**BIRADS3**	-	9	29	0	-	38	
	**BIRADS4**	-	0	0	6	-	6	
	**BIRADS 5**	-	-	-	-	-	-	
	**Total**	-	499	31	6	-	536	
**Observer 2**	**BIRADS1**	-	-	-	-	-	-	0.33
	**BIRADS2**	-	492	2	0	-	494	
	**BIRADS3**	-	10	29	0	-	39	
	**BIRADS4**	-	0	0	6	-	6	
	**BIRADS 5**	-	-	-	-	-	-	
	**Total**	-	502	31	6	-	539	
**Observer 3**	**BIRADS1**	-	-	-	-	-	-	0.23
	**BIRADS2**	-	500	2	0	-	502	
	**BIRADS3**	-	4	28	0	-	32	
	**BIRADS4**	-	0	1	6	-	7	
	**BIRADS 5**	-	-	-	-	-	-	
	**Total**	-	504	31	6	-	541	
**Left Breast**								
**Observer 1**	**BIRADS1**	-	-	-	-	-	-	0.3
	**BIRADS2**	-	408	1	0	-	409	
	**BIRADS3**	-	4	21	0	-	25	
	**BIRADS4**	-	4	0	3	-	4	
	**BIRADS 5**	-	-	-	-	-	-	
	**Total**	-	413	22	4	-	438	
**Observer 2**	**BIRADS1**	-	-	-	-	-	-	0.21
	**BIRADS2**	-	406	1	0	-	407	
	**BIRADS3**	-	3	21	0	-	24	
	**BIRADS4**	-	1	0	3	-	4	
	**BIRADS 5**	-	-	-	-	-	-	
	**Total**	-	410	22	3	-	435	
**Observer 3**	**BIRADS1**	-	-	-	-	-	-	0.24
	**BIRADS2**	-	411	1	0	-	412	
	**BIRADS3**	-	3	21	0	-	24	
	**BIRADS4**	-	1	0	3	-	4	
	**BIRADS 5**	-	-	-	-	-	-	
	**Total**	-	415	22	3	-	440	

**Table 10 diagnostics-13-01065-t010:** Interobserver agreement of BI-RADS descriptors and final assessment category in ABUS and prototype HHUS observers.

BIRADS Lexicon	κ Value(R/L)								*p* Value
	ABUS		HHUS Observers						
			Observer 1		Observer 2		Observer 3		
**Shape**	0.97	0.95	0.90	0.92	0.92	0.92	0.95	0.90	>0.05
**Orientation**	0.90	0.92	0.70	0.70	0.70	0.87	0.70	0.66	<0.05
**Margin**	0.95	0.84	0.84	0.75	0.93	0.80	0.93	0.82	>0.05
**Echo Pattern**	0.96	0.93	0.90	0.92	0.94	0.90	0.94	0.92	>0.05
**Posterior Acoustic**	0.88	0.95	0.86	0.89	0.85	0.94	0.85	0.92	>0.05
**Calcification**	0.92	0.83	0.84	0.81	0.90	0.76	0.81	0.80	>0.05
**BIRADS Assessment**	0.92	0.86	0.90	0.84	0.90	0.85	0.92	0.84	>0.05

R = right breast, and L = left breast.

**Table 11 diagnostics-13-01065-t011:** HHUS observers compared with ABUS (gold standard) in terms of lesion location.

HHUS Observers	ABUS (Gold Standard)						
		RUIQ	RLIQ	RLOQ	RUOQ	Total	*p* Value
**Right Breast**							
**Observer 1**	**RUIQ**	135	0	0	21	156	0.12
	**RLIQ**	1	76	0	0	77	
	**RLOQ**	0	0	136	1	137	
	**RUOQ**	6	0	1	159	166	
	**Total**	142	76	137	181	536	
**Observer 2**	**RUIQ**	135	3	1	21	160	0.11
	**RLIQ**	0	75	1	0	76	
	**RLOQ**	0	0	133	0	133	
	**RUOQ**	5	0	1	164	170	
	**Total**	140	78	136	185	539	
**Observer 3**	**RUIQ**	140	0	0	19	159	0.22
	**RLIQ**	1	75	0	0	76	
	**RLOQ**	0	1	137	1	139	
	**RUOQ**	1	0	1	165	167	
	**Total**	142	76	138	185	541	
**Left Breast**		**LUOQ**	**LLOQ**	**LIQ**	**LUIQ**	**Total**	***p* Value**
**Observer 1**	**LUOQ**	125	0	0	10	135	0.06
	**LLOQ**	0	114	3	0	117	
	**LLIQ**	0	0	97	0	97	
	**LUIQ**	5	0	0	84	89	
	**Total**	130	114	100	94	438	
**Observer 2**	**LUOQ**	119	0	0	8	127	0.1
	**LLOQ**	0	113	0	1	114	
	**LLIQ**	0	0	101	1	102	
	**LUIQ**	6	0	0	86	92	
	**Total**	125	113	101	96	435	
**Observer 3**	**LUOQ**	126	0	0	9	135	1.02
	**LLOQ**	0	115	0	0	115	
	**LLIQ**	0	0	99	1	100	
	**LUIQ**	3	0	0	87	90	
	**Total**	129	115	99	97	440	

**Table 12 diagnostics-13-01065-t012:** Diagnostic performance of HHUS observers compared with ABUS (gold standard).

Parameters	HHUS				*p* Value
	ABUS	Observer 1	Observer 2	Observer 3	
**Sensitivity %**	99	97	97	98	0.60
**Specificity%**	97	90	93	96	
**PPV %**	96	80	90	94	0.67
**NPV %**	94	73	73	76	
**Accuracy %**	97	95	95	97	0.67
**AUC**	0.99	0.98	0.97	0.97	

## Data Availability

The data sets generated during the current study are not available publicly due to domestic regulation of the institution. However, they are available upon request to the corresponding author.

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
