# Peer review of "Diagnostic Performance of Prototype Handheld Ultrasound According to the Fifth Edition of BI-RADS for Breast Ultrasound Compared with Automated Breast Ultrasound among Females with Positive Lumps"

_diagnostics, 2023, doi:10.3390/diagnostics13061065_

Round 1
Reviewer 1 Report
Based on the current status of breast cancer screening, this paper compared the results of ABUS and HHUS in 1004 lesions of 162 patients to evaluate their diagnostic performance. The results showed that there was no statistically significant difference in the interobserver variability and diagnostic performance of ABUS and HHUS in these patients according to the fifth BI-RADS edition.
In general, this paper compares the diagnostic performance of ABUS and HHUS in breast cancer screening, affirms the use of ABUS in screening, and also points out the shortcomings of ABUS. However, there are many problems such as the format of the paper, which need to be carefully checked and corrected. Moreover, since it is an examination of new breast cancer screening methods, the quality of the article would be improved if it could be supplemented with thoughts and prospects of new technologies in the future.
1, it is best to set the Significant result (P<0.05) of different tables in the same place and in the same format.
2, in the title part of Table 8, there is an extra space after the last word, and Table 9 reverses the order of Table 10. In addition, the format of the subheading Sensitivity, Specificity, and Receiver Operating Characteristic Analysis seem to be faulty, as I did not find a title of this format in the previous text.
3, the Discussion title was placed in the last line of the note in Figure 3 on page 16 in the same format as the note, which is obviously inappropriate.
Reviewer 2 Report
The authors compared the diagnostic performance of a prototype handheld ultrasound from Fujikin and an automated breast ultrasound (Invenia from GE) in a total of 1004 lesions from 162 participants at the Centre of Diagnostic Nuclear Imaging, Universiti Putra Malaysia according to the fifth edition of Breast Imaging Reporting and Data System (BI-RADS) lexicon. They found that there are no significant differences between these two devices in diagnostic accuracy, sensitivity, specificity, positive predictive value, negative predictive value, and areas under the curve. The results are useful for clinical sonographer and radiologists.
Abstract
Full name of HHUS should be shown in its first use.
Introduction
The location of “Centre of Diagnostic Nuclear Imaging” should be included in the first use.
Materials and Methods
Page 3 Line 12-13 grammar error
Line 22 change to “are blind to”
Results
Table 1 change “Chines” to “Chinese”
Table 3 the font size can be reduced to show the whole term in one row for easy reading
Page 7 Line 1 change “Both” to “both”
Line 12 change “17were” to “17 were”
Tables 5,7,8,9 why there is a blank row?
Figure 2 the red circle in the first low is in the wrong position, there is no explanation of the red circles, and all labels are not shown correctly
Round 2
Reviewer 1 Report
I have no other comments